# A Meta-Analysis of Bacterial Communities in Food Processing Facilities: Driving Forces for Assembly of Core and Accessory Microbiomes across Different Food Commodities

**DOI:** 10.3390/microorganisms11061575

**Published:** 2023-06-14

**Authors:** Zhaohui S. Xu, Tingting Ju, Xianqin Yang, Michael Gänzle

**Affiliations:** 1Department of Agricultural, Food and Nutritional Science, University of Alberta, Edmonton, AB T6G 2P5, Canada; zhaohui5@ualberta.ca (Z.S.X.); tju1@ualberta.ca (T.J.); 2Lacombe Research and Development Centre, Agriculture and Agri-Food Canada, Lacombe, AB T4L 1W1, Canada; xianqin.yang@agr.gc.ca

**Keywords:** nutrient availability, *Listeria monocytogenes*, *E. coli* O157, surface-associated microbiome, biofilm, food processing facility, food spoilage

## Abstract

Microbial spoilage is a major cause of food waste. Microbial spoilage is dependent on the contamination of food from the raw materials or from microbial communities residing in food processing facilities, often as bacterial biofilms. However, limited research has been conducted on the persistence of non-pathogenic spoilage communities in food processing facilities, or whether the bacterial communities differ among food commodities and vary with nutrient availability. To address these gaps, this review re-analyzed data from 39 studies from various food facilities processing cheese (n = 8), fresh meat (n = 16), seafood (n = 7), fresh produce (n = 5) and ready-to-eat products (RTE; n = 3). A core surface-associated microbiome was identified across all food commodities, including *Pseudomonas*, *Acinetobacter*, *Staphylococcus*, *Psychrobacter*, *Stenotrophomonas*, *Serratia* and *Microbacterium*. Commodity-specific communities were additionally present in all food commodities except RTE foods. The nutrient level on food environment surfaces overall tended to impact the composition of the bacterial community, especially when comparing high-nutrient food contact surfaces to floors with an unknown nutrient level. In addition, the compositions of bacterial communities in biofilms residing in high-nutrient surfaces were significantly different from those of low-nutrient surfaces. Collectively, these findings contribute to a better understanding of the microbial ecology of food processing environments, the development of targeted antimicrobial interventions and ultimately the reduction of food waste and food insecurity and the promotion of food sustainability.

## 1. Introduction

According to the Food and Agricultural Organization, the demand for food is expected to increase by 56% to meet the needs of the growing global population [1]. To address this challenge, several solutions have been proposed, with reducing food loss and waste being the most crucial one. Unfortunately, approximately 25% of food that is produced for human consumption is wasted, and this loss occurs at various stages of the food supply chain, from production to consumption [2]. One of the most significant contributors to food waste is microbial spoilage. This issue is of concern for food security, as food spoilage can lead to decreased food availability and increased prices, making it even more challenging for food-insecure populations to access sufficient nutritious food. Spoilage also is of concern for the sustainability of food production and emission of greenhouse gases, particularly for meat and meat products, which have a large ecological footprint.

The microbial spoilage of food has been widely probed within food processing facilities across various food commodities. Processing facilities serve both as an establishment niche, where they allow autochthonous microbes to colonize and persist over long periods of time, and as a persistent niche for microbes that are transmitted from raw materials or the environment [3]. Researchers have approached microbial-mediated food spoilage from different angles, by examining the microbiomes of raw materials and final products, the transmission from poor handling practices and the role of environmental surfaces during production. The assembly of bacterial communities in food processing facilities is influenced by four processes: selection, drift, speciation and dispersal [4]. More precisely, microbes entering food processing facilities are limited by dispersal, while persisting organisms are limited by selection due to routine sanitization. Species drift can be observed as the bacterial population changes in conformity with changing processing environments. For example, psychrotrophic or psychrophilic organisms such as *Pseudomonas*, *Enterobacteriaceae* and lactic acid bacteria can grow on fresh meat [5,6], seafood [6] and fresh produce [7] during cold-temperature storage, with different taxonomic abundance at the end of the shelf life. The speciation process can occur more rapidly during biofilm formation in the processing environment, as a biofilm provides an optimal environment for the exchange of genetic material horizontally and for the evolution of vertically transmitted generic material [8].

The ability of bacteria to form biofilms contributes to recurring contamination from environmental surfaces, both in food contact and non-food-contact areas [9,10,11]. In fact, the biofilm matrix serves as a physical shelter for bacterial cells, thus protecting them against antimicrobial interventions and serving as a reservoir for both spoilage and pathogenic microorganisms. Many food isolates from equipment surfaces have been shown to attach to different food materials and form biofilms in vitro [12,13]. Recent studies have also identified residential bacterial communities embedded in the biofilm matrix, as evidenced by quantifying the biofilm biomass [8,9,10]. This poses a major challenge to conventional cleaning and sanitizing procedures in food processing facilities, raising concerns about cross-contamination from processing environments to products and leading to food poisoning and deterioration issues.

The food industry, therefore, employs multiple measures to control the microorganisms arriving with raw materials and those persisting in processing facilities. In general, the Hazard Analysis Critical Control Point (HACCP) has been well adopted worldwide, primarily focusing on enhancing food safety. Decontamination of raw materials prior to production, such as washing and sanitizing fresh produce with chlorine water [14], washing animal carcasses with hot water [15,16] and monitoring animal health and farm hygiene [17], has been reported to reduce the populations of bacteria on raw materials. The efficacy of cleaning and sanitation protocols, however, depends on the hygienic design of facilities and equipment and on the training of personnel. In addition, plants are colonized by bacterial endophytes including *Bacillus*, *Burkholderia*, *Rahnella*, *Pseudomonas* and *Klebsiella* [18,19,20], which cannot be removed without heating the entire plant tissue.

Although numerous studies have identified the compositions of bacterial communities in facilities processing different food commodities and investigated the impact of processing environments on bacterial communities, little is known about how nutrient availability shapes their composition, or whether food processing facilities processing different food commodities harbor comparable bacterial communities. The compositions of bacterial communities between environmental surfaces and food products have been compared for one minimally processed vegetable facility, one artisan cheese facility and two meat processing facilities by analyzing the relative abundance of taxa [21,22,23,24]. These studies identified both the raw materials and the food processing environment as relevant sources of spoilage microbes. This review aimed to expand these analyses by providing a comprehensive summary of residential bacteria on the environmental surfaces of processing facilities of multiple food commodities and by analyzing whether the nutrient availability on specific surfaces impacts the compositions of bacterial communities.

## 2. Data Collection

Data collection was started by accessing references from four review papers [25,26,27,28] on the bacterial ecology and communities involved in food spoilage, as well as later publications that cited these reviews. Taken together, these four review papers provide a comprehensive summary of the microbial communities found in different food processing facilities, as a “beacon” for subsequent searches in citation databases. Additional studies were identified by using the keywords “food industry”, “bacterial ecology”, “bacterial communities” and “food spoilage” on Google Scholar. Priority was given to studies that sampled environmental surfaces and used 16S rRNA gene amplicon sequencing for analysis. Three studies using a culture-based approach to characterize bacterial communities in facilities producing meat and seafood were additionally included. The presence and absence of bacterial taxa in 96 samples collected from 39 processing facilities are compiled in Appendix A. The 39 processing facilities included 16 facilities processing fresh meat [10,11,22,24,29,30,31,32,33,34,35,36,37,38,39,40], 7 facilities processing seafood [41,42,43,44,45,46], 3 facilities processing RTE foods [9,47,48], 5 facilities processing fresh produce [49,50,51,52,53] and 8 facilities processing cheese [23,54,55,56,57,58,59,60]. In total, this review summarized the microbial communities from 96 environmental surfaces. Three of the 39 facilities were characterized with respect to biofilm communities; these included two meat processing facilities and one cheese facility, for a total of 13 surfaces that harbored biofilms.

Of the food processing facilities included in this study (Appendix A), RTE and cheese processing facilities were located in North America and Europe; the meat processing facilities were located in North America, Oceania and Europe. All of the seafood processing facilities sampled were located in Europe. The fresh produce processing facilities were located in North America and Asia. These geographical differences may reflect variations in processing methods, regional microbiota and cultural and environmental factors. For example, traditional and minimally processed foods (raw milk cheese and fermented meats) are favored in the European Union, whereas Americans tend to be more open to the use of technologies during production, such as the use of the hormones/antibiotics for cattle and irradiation treatment for food [61,62]. Additionally, grass-fed cattle with different breeds, shapes and sizes, processed in smaller and artisan operations, are used for consumption in the EU. In contrast, in North America, feedlot-fed cattle are raised to a uniform size for large-scale industrial production [63], contributing to a distinctive gut microbiota composition [64], which in turn potentially affects the meat quality and the environmental microbiome in the processing facility. Processing facilities and meat animals in Oceania are more similar to those in the EU than those in North America [65].

The conditions and environment vary in different processing commodities. Seafood processing facilities typically maintain relatively high humidity and a temperature of 12 °C [42], which can promote the proliferation of psychrotrophic microbes. Meat processing facilities generally maintain a temperature of less than 10 °C to preserve meat products during the majority of processing stages [66], but temperatures differ between plants and even within different rooms of the same plant. In a meat abattoir, the temperature of the production room ranged from 14 °C to 25 °C, with relative humidity between 35% and 90% [29]. In a beef processing facility, the temperature in the slaughter hall, cutting room and boning room was 10–15 °C, 4–5 °C and 11–15 °C, respectively [32]. Fresh produce processing rooms are maintained at a temperature below 8 °C [49]. The processing room temperature for the cheese industry can vary depending on the specific type of cheese being produced and the stage of the cheese-making process. In general, cheese processing facilities maintain a higher processing temperature of over 20 °C to promote the growth of mesophilic and/or thermophilic starter cultures. A lower temperature (9 °C) with high relative humidity (75%) is maintained during the ripening stage [56]. The salt concentration may additionally shape the bacterial ecology in cheese processing facilities. For example, halotolerant *Halomonas* was only identified in cheese processing facilities, potentially resulting from the brining process. Cleaning and sanitization control bacterial contamination in food processing facilities but also contribute to high temperatures and humidity [29], serving as a potential source of cross-contamination and selective pressure for microbial communities.

The datasets were analyzed using permutational multivariate analysis of variance (PERMANOVA, 999 permutations, adonis2 function, vegan package, R v4.1.0) based on the Jaccard similarity of bacterial communities with an error probability of 5% (*p* ≤ 0.05) to determine whether areas with different nutrient densities harbored different communities of microbes. The data were visualized by principal coordinate analysis (PCoA). Pairwise comparisons between groups were tested by the ‘pairwise.adonis’ function (pairwiseAdonis package, v0.4.1) with Bonferroni adjustment for multiple comparisons. Data were additionally analyzed with multiple correspondence analysis (MCA), which uses the presence of individual genera as input variables to visualize the dataset. Results of PCoA are shown in the manuscript and results obtained with MCA are provided as Appendix A.

## 3. Impact of Nutrient Source and Commodity on the Compositions of Bacterial Communities

We classified direct food contact surfaces and floor drains as “high-nutrient” areas, as these areas are characterized by the presence of product residue during processing. Non-food-contact surfaces, walls and water hoses were characterized as “low-nutrient” areas because they are unlikely to provide organic matter to support bacterial growth. The nutrient levels of floors were categorized as “unknown”. This differentiation does not account for the type of substrate (lipids, carbohydrate/sugars or proteins/amino acids), and the types of nutrients can only partially be inferred from the type of product that is processed in the specific facilities. Overall, the compositions of bacterial communities in sites with different nutrient availabilities differ (*p* < 0.05) (Figure 1). The bacterial community in high-nutrient surfaces differs from that in unknown surfaces (*p* = 0.036). Plotting the data separately by commodity revealed a partial overlap in the compositions of bacterial communities in sites with high, low and unknown nutrient availability (Appendix A), with the exception of cheese processing facilities, where high- and low-nutrient surfaces differed significantly (*p* < 0.05). The similarity of the bacterial composition between different nutrient levels within one food commodity may be attributed to the smaller sample size of sites with low or unknown nutrient availability. In contrast, MCA visualized a largely distinct composition of bacterial communities in sites with different nutrient availability with the individual taxon as input (Appendix A).

The PCoA plot of samples categorized by commodity also showed the partial overlap of the bacterial communities in facilities producing different commodities (Figure 2). Bacteria residing in RTE processing facilities shared a substantial number of bacterial taxa with other food processing facilities, while all other categories were significantly different from each other (*p* < 0.05) (Figure 2). The size of the dataset allowed further categorization by commodity and nutrient level (Figure 3). With the exception of RTE processing facilities, high-nutrient level surfaces of processing facilities exhibited distinct bacterial communities (Figure 3A). The overlap of bacteria was greater in low-nutrient sites, where only cheese plants had a significantly distinct ecology compared to meat and fresh produce processing facilities (Figure 3B). Sites with unknown nutrient density, i.e., floors, were only sampled in fresh produce, cheese and fresh meat facilities. The limited sample size perhaps largely resulted in the overlap, while the MCA plot further revealed that different commodities were clustered completely separately (Appendix A). The PCoA plot for those samples for which in situ biofilm formation was confirmed by quantification of the extracellular matrix is shown in Figure 4. The compositions of the bacterial biofilm communities in low-nutrient and high-nutrient samples were significantly different (*p* < 0.05) (Figure 4).

## 4. Which Bacteria Are Where?

Heatmaps depicting the percentages of samples in which specific taxa were present are shown in Figure 5 and Figure 6. The heatmaps were scaled to show the number of samples that tested positive for a specific taxon divided by the total number of samples. The majority of taxa depicted in the heatmaps were identified at the genus level, but some provided only family-level identification. The heatmaps shown in Figure 5 and Figure 6 differentiate samples by nutrient level and commodity, respectively. Overall, *Pseudomonas*, *Stenotrophomonas*, *Acinetobacter*, *Serratia*, *Microbacterium*, *Psychrobacter* and *Staphylococcus* were frequently present regardless of the food commodity, with *Pseudomonas* species as the most prevalent taxa (Figure 5). Meanwhile, the compositions of the bacterial communities also differed among facilities processing different food commodities.

In cheese processing facilities, *Pseudomonas* was present on 17 out of 22 environmental surfaces, followed by *Brevibacterium* and other *Bacillota*, such as *Staphylococcus*, *Lactobacillus*, *Streptococcus* and *Lactococcus* (Figure 5). Because most studies used in this meta-analysis identified bacteria at the genus level and were completed before the taxonomic re-organization of the genus *Lactobacillus* in 2020 [67], *Lactobacillaceae* are often identified at the family level only (Figure 5 and Figure 6); this communication uses the current taxonomy where this is supported by the data and the term “*Lactobacillaceae*” or “lactobacilli” otherwise. The processing steps in cheese production impact the compositions of bacterial communities. For instance, brining and the use of surface ripening provide favorable conditions for the growth of acid-sensitive, salt-tolerant and psychrotrophic bacteria, which were abundant on smear-ripened cheeses but were also identified on environmental surfaces [57,58]. Coryneforms, such as *Brevibacterium* and *Corynebacterium*, as well as *Halomonas* and *Staphylococcus* were among the main microbial genera that were identified on the surfaces of smear-ripened cheeses [68]. These organisms may cause defects in other types of cheese [69]. The high prevalence of *Lactobacillus*, *Streptococcus* and *Lactococcus* on surfaces is unsurprising given their roles as starter cultures for cheese production [23,57]. The *Lactobacillus* species detected were *L. delbrueckii* and *L. helveticus*, originating from thermophilic starter cultures used in cheese making. Equipment surfaces primarily harbored Gammaproteobacteria such as *Psychrobacter*, *Acinetobacter* and *Pseudoalteromonas*, which can cross-contaminate food samples [23,57,58]. The origin of the microbiome on surfaces in cheese processing facilities varies among different plants and remains unclear. For example, *Corynebacterium*, *Staphylococcus* and *Sphingobacterium* can be part of the raw milk or human skin microbiota [70] and subsequently spread to equipment surfaces. Lactose carry-over from vat milk or whey to non-food-contact surfaces may contribute to the higher abundance of *Staphylococcus* spp. in cheeses compared to other commodities, since lactose can stimulate biofilm formation by *Staphylococcus* [71].

In meat processing facilities, common food spoilage bacteria including *Pseudomonas*, *Acinetobacter* and *Psychrobacter* were identified on over one third of the environmental surface samples (Figure 5). The phylum of *Bacillota* also had a relatively high abundance with the presence of *Staphylococcus*, *Brochothrix*, *Bacillus* and *Streptococcus*. In addition to transmission from the human and animal skin microbiota, the high abundance of *Staphylococcus* and *Corynebacterium* was also detected in air samples throughout a poultry slaughtering house [39]. *Bacteroidota*, including *Chryseobacterium* and *Flavobacterium*, have the potential to cause the spoilage of meat and were isolated from both meat carcasses and environmental surfaces [33,34,35]. *Brochothrix* is recognized as a spoiler of raw and packaged meat and was identified on food processing surfaces [10,11,39]; it readily grows on meat and at low storage temperatures [72], even if the contamination from equipment surfaces begins with a low cell population. *Enterobacteriaceae* and lactic acid bacteria including lactobacilli, *Leuconostoc* and *Carnobacterium* also play important roles in meat spoilage, either as spoilage organisms or as protective microbes that inhibit spoilage by others. Vacuum-packaged fresh meat has a refrigerated shelf life of over 2 months, and which of the microbes on meat grow during storage depends on the meat composition, the presence of competing microbes, the storage conditions, the packaging methods and the oxygen availability [5]. In these products, *Enterobacteriaceae* are present in high abundance on the processing facilities’ surfaces but to a lesser extent in raw materials and products at the end of the shelf life, whereas lactic acid bacteria dominate the meat microbiota at the end of the shelf life, with low abundance in both processing surfaces and raw materials [24]. Psychrotrophic clostridia, mainly *Clostridium estertheticum*, cause blown pack spoilage. While the studies reviewed in this article did not identify the presence of psychrotrophic clostridia, these bacteria are known to be prevalent in the pelts and feces of slaughtered animals and have been detected in meat slaughtering facilities through the PCR amplification of specific 16S rRNA regions [73]. *Enterobacteriaceae* such as *Serratia*, *Enterobacter* and *Hafnia* have also been linked to blown pack spoilage. In the 39 studies analyzed in the current study, *Serratia* and *Enterobacter* were more frequently identified than *Hafnia* (Figure 5).

The bacterial communities in RTE processing facilities did not exhibit significant variations compared to other food commodities (Figure 2), given the processing of diverse raw materials for the respective products. Despite variations in the bacterial community across three RTE processing facilities, members of the genus *Pseudomonas* have been consistently found on different environmental surfaces, including slicers, walls and other food contact surfaces [9,47,48]. Their persistence even after regular sanitization protocols results from biofilm formation on abiotic surfaces, which may serve as an indicator of the efficacy of cleaning and sanitization practices to eradicate biofilms in food processing facilities. Other spoilage-related taxa, such as *Enterobacteriaceae, Streptococcaceae*, lactobacilli, *Brochothrix* and *Leuconostoc*, have been found to colonize on equipment surfaces and to occur on RTE food products [47]. Moreover, lactic acid bacteria, especially *Leuconostoc* spp., grow at refrigeration temperatures and typically dominate RTE meat microbiota at the end of the shelf life [5].

The food contact surfaces of seafood processing facilities were characterized by the unique presence of *Glutamicibacter*, *Aliivibrio*, *Escherichia*, *Morganella*. *Glutamicibacter* and *Morganella*, which are associated with ocean fish [44,46]. *Morganella* is a copious producer of histamine during the storage of seafood, which can lead to intoxication after the consumption of seafood, particularly scombroid fish [74]. In addition, common seafood spoilers identified among diverse seafood products, such as *Pseudomonas*, *Acinetobacter*, *Serratia*, *Psychrobacter* and *Brochothrix*, have also been isolated from environmental surfaces, suggesting the possibility of contamination from environmental surfaces. Marine spoilage bacteria including *Aeromonas*, *Pseudoalteromonas*, *Photobacterium* and *Shewanella* are mostly found in marine systems and seafood samples, contributing to seafood off-flavors and limited shelf-lives. An analysis of a salmon processing facility revealed the presence of *Aeromonas* and *Shewanella* on environmental surfaces and in seawater, serving as a source of contamination of salmon fillets [44]. On the other hand, *Pseudoalteromonas* and *Photobacterium* were absent on environmental surfaces but were found in raw fish and seawater [14]. Lactic acid bacteria, particularly *Carnobacterium* spp., have been isolated from fish guts and aquatic environments [75]. In both meat and seafood products, the growth and metabolism of *Carnobacterium* spp. during refrigerated storage can have beneficial or detrimental effects on product quality; this depends on the strain- or species-specific metabolic traits [76,77]. Moreover, the nutrient availability also shapes the microbial composition in seafood processing facilities. For example, the genera *Aeromonas*, *Acinetobacter*, *Pseudomonas*, *Shewanella*, *Chryseobacterium* and *Flavobacterium* were present on both high- and low-nutrient surfaces, while *Comamonas* was exclusively found on low-nutrient surfaces. Common ecological niches for *Comamonas* include freshwater, wastewater, the fish gut and plants [78,79].

In fresh produce facilities, the most commonly identified genera were *Pseudomonas* and *Acinetobacter* from food contact surfaces and *Comamonas*, *Chryseobacterium* and *Janthinobacterium* from non-food-contact surfaces, such as trolley and floor drains [49]. *Janthinobacterium* was abundant in freshwater and fresh vegetables such as lettuce surfaces [80], which could increase the risk of spoilage in fresh produce. In addition, it was also found that *Comamonas* and *Janthinobacterium* synergistically interacted with other microorganisms such as *Serratia* [49], contributing to the negative role in the shelf life of fresh-produce. Furthermore, fresh produce facilities uniquely harbored the plant-associated microbes *Rahnella* and *Ralstonia* [50]. A strain of *Ralstonia* spp. was confirmed as a strong biofilm producer under low-temperature conditions (<10 °C), enhancing the mixed-species biofilm formation together with *E. coli* O157:H7, *Listeria monocytogenes* and *Salmonella* [12]. Taking into account the influence of nutrient levels on the compositions of bacterial communities in the fresh produce production environment, the distinct presence of *Cellulosimicrobium*, *Corynebacterium*, *Sphingobacterium*, *Klebsiella*, *Microbacterium* and *Rahnella* was observed on nutrient-abundant surfaces across the five studies, while *Arthrobacter*, *Rhizobium*, *Rhodoferax*, *Paenibacillus* and *Staphylococcus* only occurred on nutrient-deficient surfaces. Other common soil bacterial genera such as *Cupriavidus*, *Burkholderia* and *Devosia* have been isolated from plant tissues [20,81,82] and uniquely presented in fresh produce processing facilities with relatively high occurrence (Figure 5).

## 5. Can a Core Microbiome in Food Processing Facilities Be Identified?

A core surface-associated microbiome of food processing facilities was identified from the 39 studies with the following order of taxa: *Pseudomonas*, *Acinetobacter*, *Staphylococcus*, *Psychrobacter*, *Stenotrophomonas*, *Serratia* and *Microbacterium*. These seven genera can be further characterized into two sub-groups: (i) organisms that are commonly identified as food spoilage organisms, including *Pseudomonas*, *Acinetobacter*, *Psychrobacter* and *Serratia*, and (ii) proximate microorganisms with spoilage potential. The spoilage potential of *Staphylococcus*, *Stenotrophomonas* and *Microbacterium* has been confirmed in various studies through their ability to degrade lipid and protein in vitro [83,84]. In addition, *Staphylococcus aureus* causes food poisoning through the production of enterotoxins. Outbreaks associated with *S. aureus* have occurred in various types of food and are often linked to improper handling and poor personal hygiene. Food isolates of *S. aureus* may also pose a risk of transmission of multi-drug-resistant *Staphylococcus* to humans through food consumption [85].

The core microbiome identified among different food commodities is not coincidental. Firstly, *Pseudomonas*, *Acinetobacter*, *Psychrobacter* and *Serratia* are commonly found in natural environments such as soil and water and have a versatile lifestyle, which allows them to utilize diverse energy sources and grow at lower temperatures [86,87,88,89,90]. Therefore, the commonly used method to extend the shelf life, refrigeration, does not prevent their growth. Modified atmosphere packaging is currently in use to control the growth of *Pseudomonas*, *Acinetobacter* and *Psychrobacter* based on their strictly aerobic features, while the facultative anaerobic *Serratia* spp. have been detected in the end products [86,87,88,89]. Secondly, the growth of spoilage bacteria on food is often associated with the production of volatile compounds, which is a common signal of food deterioration. Given the involvement of bacterial volatile compounds in interkingdom interactions [90,91], the volatiles may additionally act as signaling molecules that modulate the growth of other bacteria in food products and processing environments, and they may further impact the deterioration of food products, bacterial colonization and biofilm formation on food equipment surfaces. This hypothesis, if confirmed, can significantly broaden our understanding of the dynamic interactions between bacterial volatile compounds, spoilage issues and biofilm formation. Third, the core microbiome apparently resists cleaning and disinfection strategies in facilities processing different food commodities, including seafood, fresh meat, RTE and cheese [40,42,84]. Although cleaning and sanitization are not intended to achieve sterility in food processing facilities, the identification of a core microbiome that has implications for the shelf lives of products suggests that it may be necessary to implement more effective strategies to eradicate these microorganisms from food processing environments.

The differentiation of the bacterial communities in processing facilities by nutrient availability (Figure 6 and Figure 7) revealed that eight core taxa, *Arthrobacter*, *Brevibacterium*, *Flavobacterium*, *Staphylococcus*, *Pseudomonas*, *Psychrobacter*, *Stenotrophomonas* and *Enterobacter*, were shared among all three different nutrient-variable niches. Nutrient-rich areas specifically harbored 16 bacterial genera, especially with the relatively high presence of *Serratia*, while *Xanthomonas* was only present in nutrient-scarce environments (Figure 7). The adaptation of the oligotroph *Xanthomonas* to nutrient-deficient conditions has been linked to its low copy number of ribosomal RNA operons [92,93]. *Arthrobacter* is a genus of mainly soil bacteria with nutritional versatility. For example, it can utilize diverse sources as carbon and energy sources, such as carbohydrates, organic acids, amino acids, aromatic compounds and nucleic acids [94], leading to its presence on floors and surfaces, including high- and low-nutrient surfaces. *Brevibacterium* spp., mainly present in meat and cheese processing facilities, can metabolize different carbon sources, such as glucose and galactose, which are relatively abundant in meat and cheese processing facilities. *Brevibacterium* also exhibits resistance to carbohydrate starvation [95], which perhaps explains its survival under the conditions of nutrient-deficient surfaces. Knowledge of the nutrient adaptability among *Flavobacterium* species is limited. However, it displays physiological diversity, which further results in its wide distribution across different food manufacturers. Habitats include, but are not limited to, cold freshwater and aquatic environments, soil and food products such as fish, raw and processed meat, dairy products and agricultural crops [96]. The ability of *Staphylococcus*, *Pseudomonas*, *Psychrobacter* and *Stenotrophomonas* to form biofilms [97,98] allows them to reside and disperse on diverse surfaces with different nutrient levels, thus becoming frequent contaminants in food production areas. Microbial communities from high-nutrient surfaces tend to be different from those on floor surfaces. This difference was also visualized in the Venn diagram, as the microorganisms did not overlap between high-nutrient and unknown surfaces (Figure 7), while floor samples did harbor some unique microorganisms with a relatively low frequency of presence (Appendix A).

The persistence of diverse microbial communities among different processing facilities is likely related to the presence of these microbes in biofilms. Information on the strain-level (fewer than 20 SNPs) persistence of microbes in food processing facilities is available for *Listeria monocytogenes*, which is of particular concern for the food industry because it causes foodborne disease associated with the consumption of cheeses, produce and RTE meats [99]. Sampling of a few meat processing facilities for approximately one year after the start of operation revealed that the facility was colonized by strains of *Listeria* within three months and that some of these strains persisted as part of the microbiome in the facility [100]. A seafood processing facility in the U.S. harbored the same strains of *Lm. monocytogenes* over a period of 17 years, and the calibration of the mutation rates of these strains indicated that the strains likely colonized the facility after operations started in 1974 and had remained in the facility since then [101]. The typing of *E. coli* O157:H7 by pulsed-field gel electrophoresis in 21 “high event period” incidents across nine beef processing facilities throughout the United States identified strains of *E. coli* O157:H7 with the same pulsed-field gel electrophoresis patterns over extended periods of time in the same facility (two or more outbreaks in the same facility) and across facilities in the same geographical region [102]. Similar findings were noted for generic *E. coli*, which had clonal strains persisting in the same facilities contaminating cuts and trimmings, as determined by multiple-locus variable-number tandem repeat analysis [103,104]. Some of these strains were obtained after the cleaning of the non-contact surfaces of conveyor belts [105]. The clonal relationship of post-sanitation strains was further confirmed by whole-genome analysis, with a cut-off for SNPs at <20 [106]. Persistent strains of *E. coli* were also observed for *E. coli* O157:H7 on pig farms, resulting in outbreaks [107]. In addition to biofilm formation, strains of *E. coli* may achieve persistence via their ability to utilize novel substrates [107]. Pathogenic bacteria, however, are not the primary biofilm-forming organisms in food processing facilities but inhabit biofilms that are formed by other microbes [26]. Strain-level identification of the persistence of spoilage microbes is currently not available, but the presence of a core microbiota that remained unchanged for over 6 years in a meat processing facility implies the strain-level persistence of spoilage microbes as well [40].

Most common food spoilage microorganisms, including *Pseudomonas* species, exhibit a strong biofilm formation ability across various food processing environments, regardless of the nutrient availability [11,98]. Despite the extensive data available on the microbial ecology of food processing facilities and the ability of isolates to form biofilms, only a few studies have analyzed the microbial communities in biofilm samples from food processing facilities. Our study summarized 13 biofilm communities from one cheese processing industry and two meat processing facilities [11,54,108]. Three out of thirteen were from “low-nutrient” areas, collected from water hoses in meat processing facilities. Overall, the nutrient availability significantly (*p* < 0.01) impacted the biofilm bacterial communities (Figure 4). *Rhodococcus*, *Stenotrophomonas*, *Microbacterium* and *Flavobacterium* were frequently seen in samples from water hoses, a site with low nutrient levels. The former two genera were absent in high-nutrient-level surfaces (Figure 8B). *Rhodococcus* has been previously isolated from pink biofilms in bathrooms [109] and it catabolizes a variety of substrates [110], which could explain its ability to thrive in an environment with low nutrient levels. Other genera, such as *Brevundimonas*, *Janthinobacterium*, *Micrococcus*, *Paeniglutamicibacter*, *Pseudoclavibacter* and *Sphingomonas*, were only detected on low-nutrient-level surfaces. In contrast, the high abundance (>50%) of *Pseudomonas*, *Psychrobacter, Brochothrix*, *Acinetobacter*, *Lactococcus* and *Carnobacterium* was detected on nutrient-rich surfaces, and the latter two were absent on poor-nutrient surfaces (Figure 8B). In particular, *Pseudomonas* was detected in all nine nutrient-rich biofilm samples. The nutrient availability is critical for *Pseudomonas fluorescens* to switch between free living cells and biofilm-embedded cells by regulating the production of a signaling molecule, cyclic-di-GMP. Briefly, bacterial cells tend to attach to surfaces and form biofilms under high-nutrient conditions, while nutrient scarcity encourages cell dispersal with a lower level of cyclic-di-GMP [111]. In the food manufacturing setting, the nutrient availability on equipment surfaces fluctuates. On the one hand, this regulatory pattern can increase resistance to cleaning and sanitation by biofilm formation when nutrient levels are high but, on the other hand, it favors cross-contamination to other surfaces through dispersal when nutrients are scarce. Other common food spoilers such as *Shewanella*, *Staphylococcus*, *Streptococcus*, *Pseualteromonas*, *Leuconostoc* and *Kocuria* are also part of the biofilm constitution isolated from nutrient-rich surfaces such as cutters and screw conveyors (Figure 8B). The diverse microbial communities in high-nutrient surfaces were largely attributed to floor drain biofilms as drainage provides a relatively stable niche. For instance, 15 different genera were present in floor drain biofilms from meat processing facilities, while 20 different genera were isolated from floor drain biofilms in cheese processing facilities. Only *Lactococcus* and *Pseudomonas* overlapped in meat processing and cheese production facilities (Figure 8A) [11,54,108].

## 6. The Use of Sanitizers and Selective Ecology

The appropriate, hygienic design of equipment and facilities, together with cleaning and sanitization procedures and training of personnel, are the primary strategies to control resident microbes and to mitigate the risk of introducing microorganisms to food processing environments through raw materials, employees, water, soil and air. Improper hygienic design results in niches, or “dead areas”, that are difficult to access during routine maintenance and inspections and are thus difficult to clean [112]. In addition, cleaning and sanitization procedures may only be partially effective and further shape the bacterial ecology in food processing facilities via the following pathways. First, some bacteria are eliminated while other bacteria are capable of surviving such efforts and persist within the facility. For example, the genera *Janthinobacterium* and *Aeromonas* were eliminated after cleaning and sanitization practices in a beef slaughtering plant, while *Pseudomonas*, *Comamonas*, *Acinetobacter* and *Flavobacterium* were not [10]. Cleaning and sanitation may thus inadvertently encourage the growth and spread of some undesirable, resistant microorganisms that persist in the processing environment after more harmless competitors are eliminated. Second, bacteria may also acquire resistance to sanitizers due to repeated exposure to sublethal concentrations of biocides. For example, strains of *E. coli* isolated from chlorine-treated wastewater samples harbored the transmissible locus of stress tolerance genomic island, increasing its tolerance to common sanitizers in both planktonic and biofilm-embedded cells [113,114]. Achieving the desired concentration of sanitizer on equipment surfaces to effectively kill bacteria is also challenging, as the presence of water or debris on the surface can dilute the concentration of the sanitizer, while scratches and damages to equipment can serve as hidden habitats. For instance, the use of quaternary ammonium compounds, commonly used in food processing facilities to control *Listeria* spp., can promote the acquisition genes coding for resistance [115,116]. Third, the formation of biofilms on surfaces provides a physical barrier that limits diffusion and results in low levels of exposure to sanitizers among bacteria in the interior of the biofilm [8]. A higher proportion of biofilm-embedded cells survived after continuous exposure to benzalkonium chloride when compared to planktonic cells of *Salmonella* Enteritidis [117]. The formation of biofilms on surfaces in food processing facilities represents thus a survival strategy [118] to adapt to the harsh conditions, including hot steam, large temperature changes and oxidative stress. Lastly, cleaning and sanitization contribute to high temperatures and humidity, thus favoring bacterial growth, and may promote cross-contamination. For instance, the most abundant bacterial genera recovered from a seafood processing facility after cleaning and sanitization belonged to *Aerococcus*, *Serratia*, *Enterobacter*, *Kocuria*, *Citrobacter*, *Pseudomonas* and *Acinetobacter*, and the latter three were identified as strong biofilm producers at low temperatures [42]. These findings thus further underscore the need for effective cleaning and sanitization in food processing facilities.

## 7. Limitations

This study highlights how nutrient availability and the processing of different food commodities shape the compositions of surface-associated microbial communities in common food processing facilities. Many bacterial activities and characteristics are strain-dependent, and compiling information mainly at the genus level may not fully capture the variations in each individual strain. A focus on strain-level characterization could provide a more comprehensive understanding of the microbial communities in food processing environments. Additionally, the relative abundance of associated microorganisms was not considered here, as most studies considered only the presence of specific taxa, while information on abundance was often missing.

More studies are focused on microbial communities in meat processing and cheese processing facilities, potentially leading to biases and confounding, which may have impacted the conclusions on food commodities with fewer data points, such as seafood processing facilities. Microbial communities that are associated with low-nutrient and unknown sites, such as non-food-contact surfaces and floors, are also often sampled less frequently. However, the accumulation of physical, chemical and biological hazards on non-food-contact surfaces and floors can cause cross-contamination to food contact surfaces. In addition, sanitation efforts typically focus less on non-food-contact surfaces and floors compared to food contact surfaces. Therefore, future studies should consider sampling more areas, such as non-food-contact surfaces and floors, to better understand microbial dispersal within facilities and ultimately help food processing facilities to develop more comprehensive sanitation protocols. In addition, facilities processing other perishable products, such as eggs and milk, were not included.

## 8. Conclusions and Perspectives

Our meta-analysis of the microbial communities in food processing facilities indicates that the composition of the bacterial community differs when exposed to different nutrient levels in the food manufacturing environment. The influence of nutrient availability on the bacterial community is even more pronounced in biofilm-embedded cells. In addition, we identified a core community across food processing facilities irrespective of the commodity that was processed, as well as accessory microbiomes associated with specific food commodities.

In ecological terms, processing facilities represent an establishment niche [3] for autochthonous microbes that colonize food processing facilities over evolutionarily relevant timelines. The composition of these microbial communities is mainly shaped by selection and speciation. Processing facilities also represent a persistence niche [3] for allochthonous microbes, which establish a temporary and not a permanent presence. The composition of these microbial communities is shaped by selection and dispersal limitations [4].

The control of allochthonous microbes relies on the control of dispersal by personnel, air and water and by the control of microbes that are associated with the raw materials. Animals and plants, however, are invariably associated with commensal microbiota that will enter facilities that process fresh meat or plants. Autochthonous microbes reside on non-food-contact surfaces, where they are not eliminated by routine sanitization measures. Dispersal from these non-food-contact surfaces to food is mediated by factors such as condensation, airflow and drain back-ups. Cleaning and sanitization can contribute to dispersal, e.g., by high-pressure washing that generates aerosols [10,119]. Both allochthones and autochthones are impacted by improvements in the hygienic design of processing facilities and equipment, improved cleaning and sanitization protocols and the improved training of personnel in food safety management. Our meta-analysis also underlines that more studies are required to explore the hidden activities of bacteria on non-food-contact surfaces (hidden areas) and to study biofilms as polymicrobial communities in food processing plants. The reconstitution of these polymicrobial biofilms in vitro would allow us to probe the distribution of each bacterium in this complex microbial system.

Indisputably, food waste due to microbial spoilage is closely connected to the environment, animal feed and human consumption. We thereby propose the concept of “one sustainability” to complement the “one biofilm, one health” concept [120], to emphasize the importance of reducing food waste and promoting sustainability in the food industry, which could help to ensure that food resources are used more efficiently and that more people have access to safe and nutritious food.

## Figures and Tables

**Figure 1 microorganisms-11-01575-f001:**
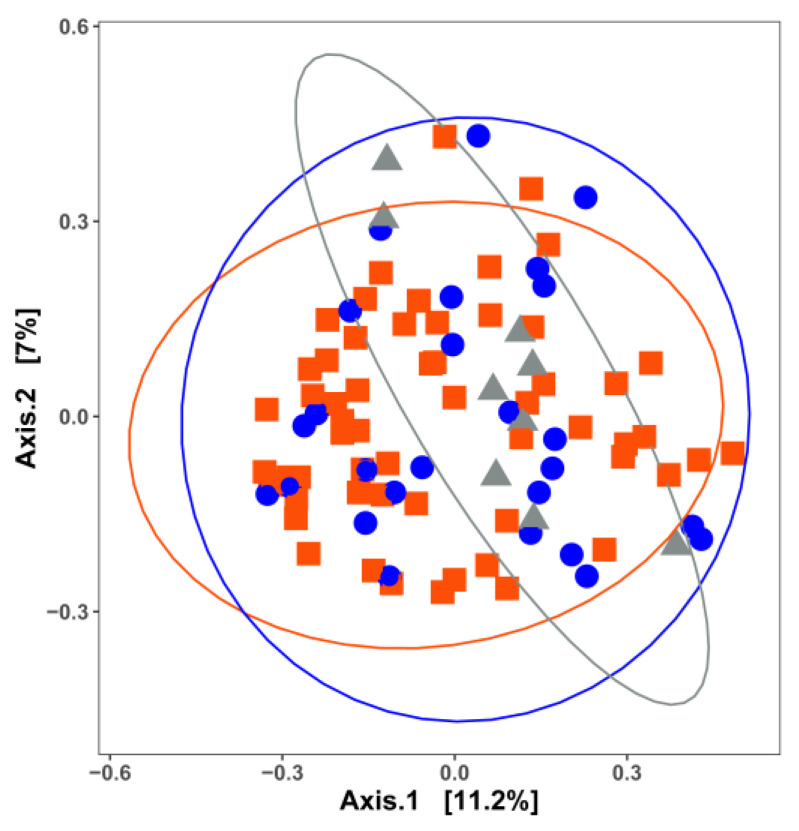
Principal coordinate analysis (PCoA) based on the Jaccard distance matrix for 96 surface-associated samples with different nutrient availabilities. Samples are colored by nutrient availability: red, high nutrient; blue, low nutrient; grey, unknown surfaces. Permutational multivariate analysis of variance was used to statistically differentiate among the bacterial communities. Bacterial communities are significantly different (*p* = 0.047) among surfaces with different nutrient availability. Bacterial communities on surfaces with unknown nutrient availability tend to differ from those on high-nutrient surfaces (pairwise-adjusted *p* = 0.084).

**Figure 2 microorganisms-11-01575-f002:**
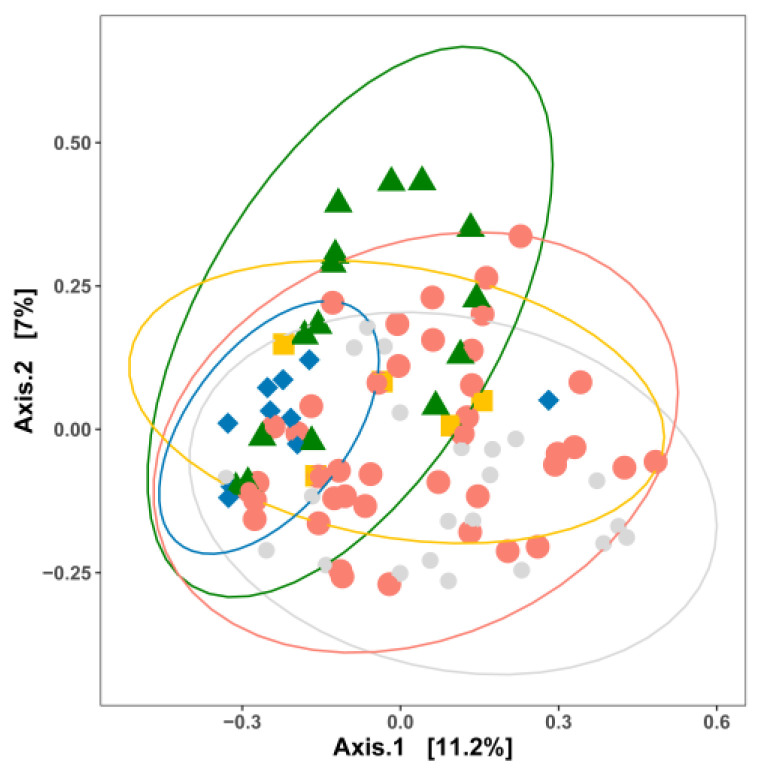
Principal coordinate analysis (PCoA) based on the Jaccard distance matrix for 96 surface-associated samples from different food commodities. Samples are colored by food commodity: yellow, RTE processing facilities; red, meat processing facilities; blue, seafood processing facilities; green, fresh produce processing facilities; light grey, cheese processing facilities. Permutational multivariate analysis of variance was used to statistically differentiate among bacterial communities. The associations of community variance with different food commodities are displayed in Appendix A.

**Figure 3 microorganisms-11-01575-f003:**
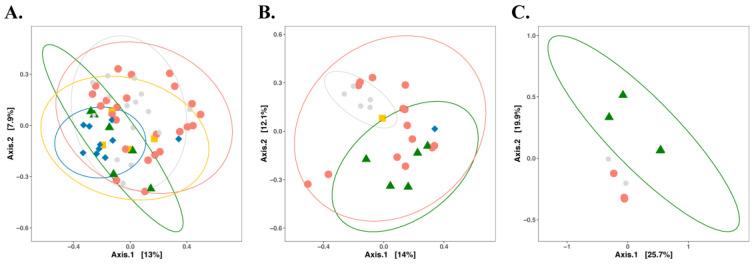
Principal coordinate analysis (PCoA) with Jaccard index for bacterial diversity based on 5 food processing facilities associated with different nutrient levels. Points represent microbial communities collected from different processing facilities and are clustered based on the same nutrient level: (**A**) high; (**B**) low; (**C**) unknown. Light grey, cheese processing facilities; green, fresh produce processing facilities; red, meat processing facilities; yellow, RTE processing facilities; blue, seafood processing facilities. Permutational multivariate analysis of variance was used to statistically differentiate among bacterial communities. The associations of community variance with different food commodities for high- and low-nutrient surfaces are displayed in Appendix A, respectively.

**Figure 4 microorganisms-11-01575-f004:**
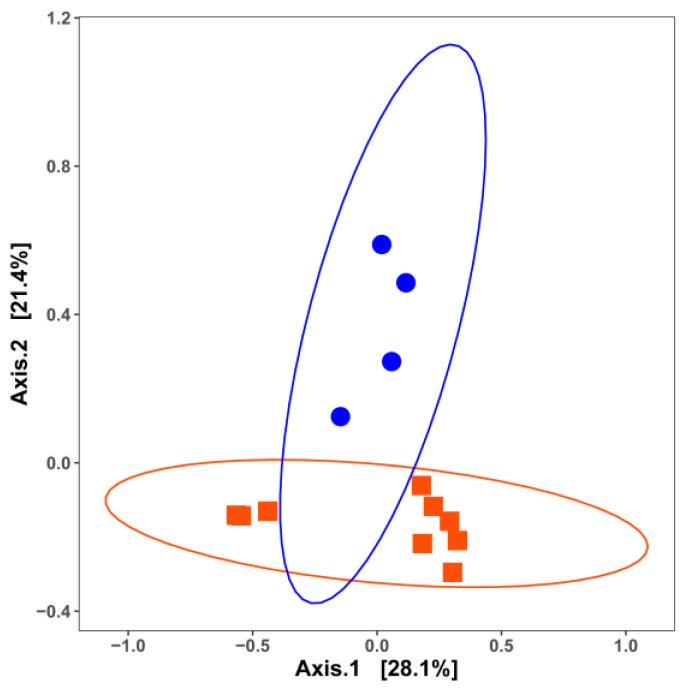
Principal coordinate analysis (PCoA) plot with Jaccard similarity for bacterial diversity among environmental biofilms formed under different nutrient levels: red, high nutrient; blue, low nutrient. Data collected from two meat processing facilities and one cheese processing facility, contributing to 13 sampling surfaces in total. Permutational multivariate analysis of variance was used to statistically differentiate among bacterial communities.

**Figure 5 microorganisms-11-01575-f005:**
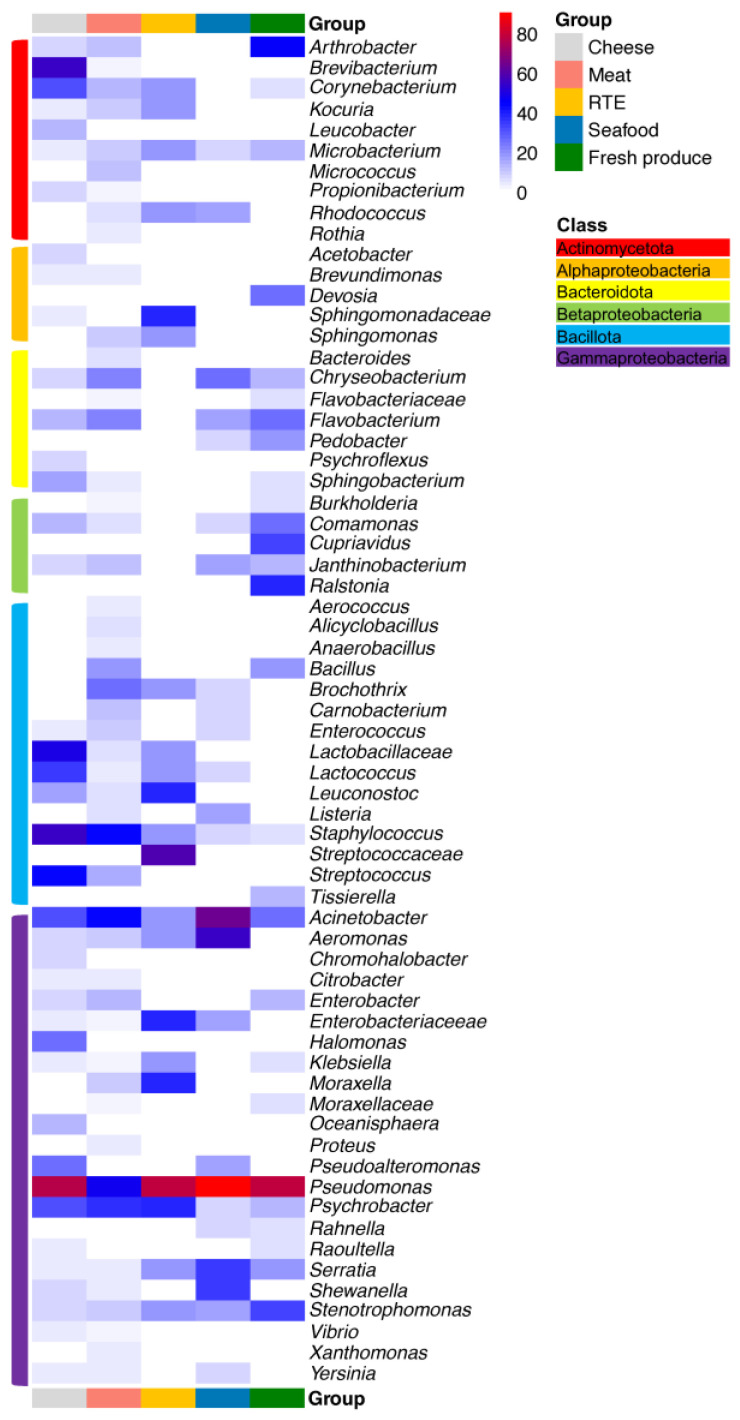
Heatmap depicting the relative abundance of the occurrence of bacterial genera present in different food processing facilities.

**Figure 6 microorganisms-11-01575-f006:**
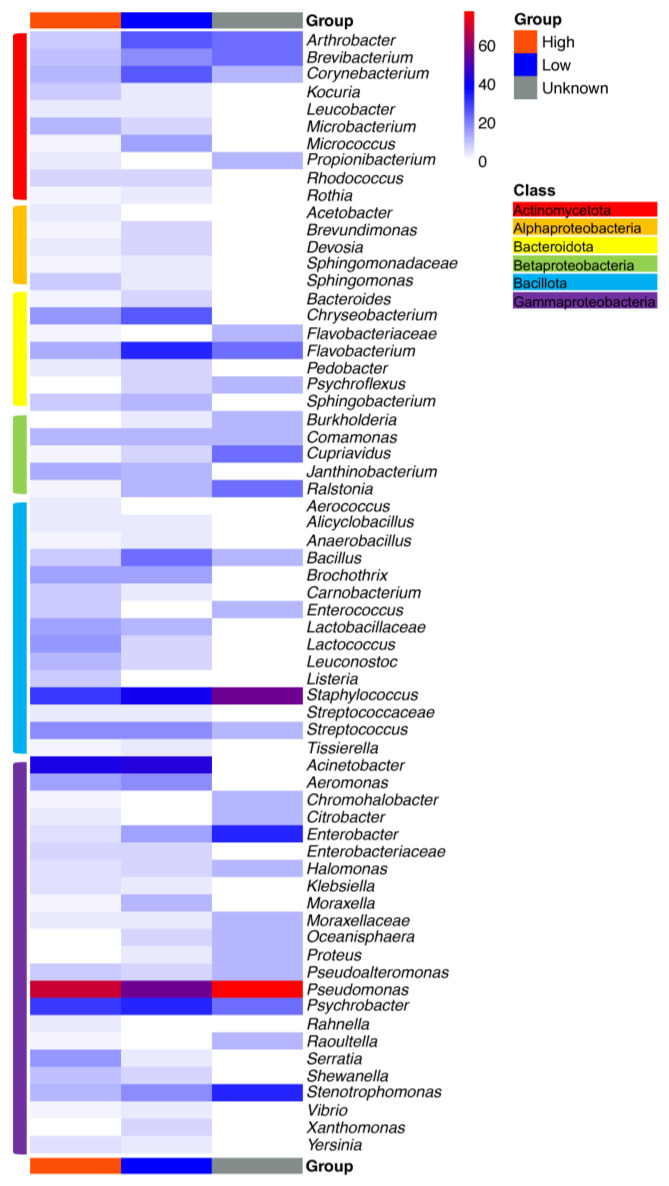
Heatmap depicting the relative abundance of the occurrence of bacterial genera present in samples with different nutrient levels.

**Figure 7 microorganisms-11-01575-f007:**
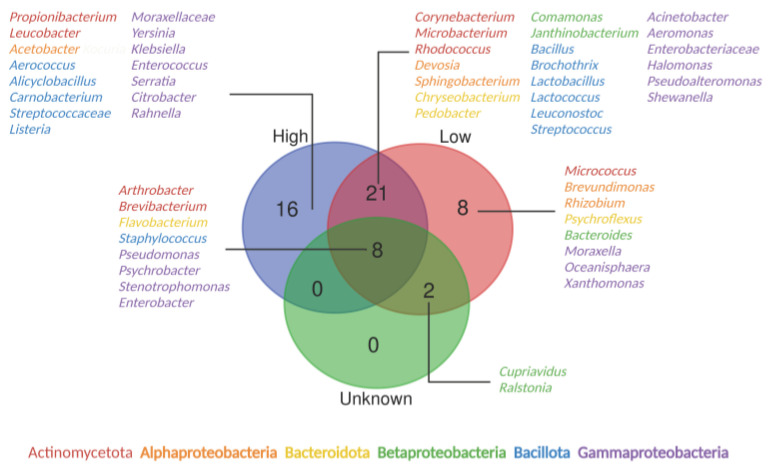
Venn diagram presenting the genera shared among environmental surfaces with different nutrient levels. Blue, high nutrient; red, low nutrient; green, unknown.

**Figure 8 microorganisms-11-01575-f008:**
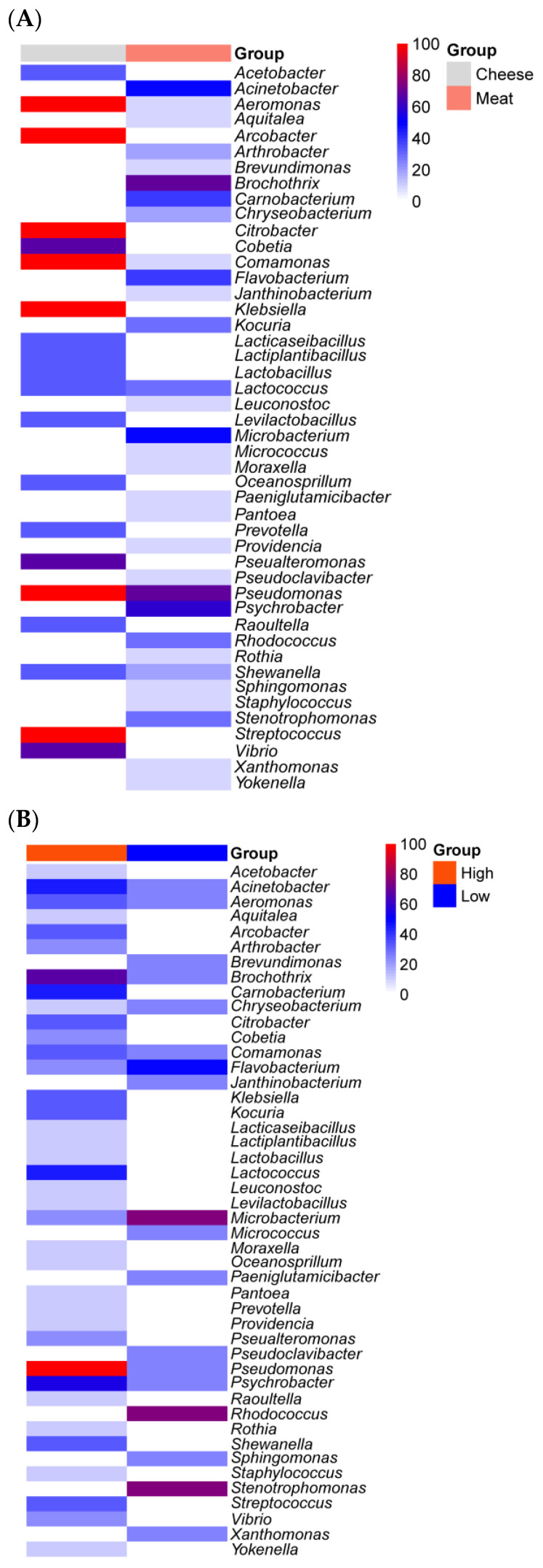
The relative abundance of bacterial genera residing in biofilm consortia based on different food categories (**A**) and nutrient levels (**B**). Data collected from two meat processing facilities and one cheese processing facility, contributing to 13 sampling surfaces in total.

## Data Availability

All data used in the study are contained in the manuscript or in the cited publications.

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
