# Peer review of "A Meta-Analysis of Bacterial Communities in Food Processing Facilities: Driving Forces for Assembly of Core and Accessory Microbiomes across Different Food Commodities"

_microorganisms, 2023, doi:10.3390/microorganisms11061575_

Round 1
Reviewer 1 Report
The paper highlights the effect of nutrient availability and processing of different types of food (meat, cheese, seafood, fresh produce, and ready-to-eat products) on the composition of microbial communities during food processing facilities, by a consistent and good meta-analysis. The manuscript is very interesting. The review article is well presented and performs an important and current literature search.
Minor comments:
Please, the Authors could control the correct genera name of bacteria in figures and in the text (i.e. fig.5, Lactobacillus; P7, L246, Lactobacillus; fig. 6; etc..)
The Authors considered nutrient densities or intensity in the different analyses. For example, in Figure 1 or S1. What type of nutrient? Explain please.
When did the Authors define “high” and “low” nutrient, what it means? Did it do a characterization or a quantification? Did you hypothesize nutrient distribution in food processing facilities?
P5, L189-190: High-nutrient level surfaces of processing facilities excluding RTE processing facilities exhibited distinct bacterial communities.
Could you try to explain this sequence?
Fig. 5 and 6: Is it possible to add a samples legend also at the bottom of the figure? The "color group" is on the top of the figure, but I believe that if "color group" will be also added on the bottom, the figure will be easy to see/understand. In the end, is it possible to invert the image? In a vertical sense.
Reviewer 2 Report
A very interesting manuscript that presents a meta-analysis of bacterial communities in food processing facilities. The manuscript is very well written and loaded with useful information. A couple of typos detected (l. 11, 23) do not reduce its the quality, it can be published as submitted.
Reviewer 3 Report
It was my pleasure to review the manuscript on the meta-analysis of bacterial communities in food processing facilities by Xy et al.
Some of my minor comments are listed below:
Line 78. "Animal washing with hot water" - difficult to imagine. Could this be clarified somehow?
Line 98. Why were those four papers selected for the initial screening? Could the authors add an explanation?
Line 121. "Hormones/antibiotics for cattle". This cannot be considered a novel technology
Line 131. The temperature inside the meat processing facilities usually is not strictly regulated except for cutting/deboning rooms.
Line 252. The origin of the biome remains unclear?
Line 314 -315. Positive and negative roles. The sentence is unclear.
Line 443. "Our study ..." Could the author consider moving the biofilm part to a separate subsection or paragraph?
The conclusion section could be shortened.
